# Hemp (*Cannabis sativa* L.) Flour-Based Wheat Bread as Fortified Bakery Product

**DOI:** 10.3390/plants10081558

**Published:** 2021-07-29

**Authors:** Iulian Eugen Rusu, Romina Alina Marc (Vlaic), Crina Carmen Mureşan, Andruţa Elena Mureşan, Vlad Mureşan, Carmen Rodica Pop, Maria Simona Chiş, Simona Maria Man, Miuţa Rafila Filip, Bogdan-Mihai Onica, Ersilia Alexa, Vasile-Gheorghe Vişan, Sevastiţa Muste

**Affiliations:** 1Food Engineering Department, Faculty of Food Science and Technology, University of Agricultural Sciences and Veterinary Medicine Cluj-Napoca, 3-5 Calea Manaştur Street, 400372 Cluj-Napoca, Romania; iulian.rusu@usamvcluj.ro (I.E.R.); andruta.muresan@usamvcluj.ro (A.E.M.); vlad.muresan@usamvcluj.ro (V.M.); simona.chis@usamvcluj.ro (M.S.C.); simona.man@usamvcluj.ro (S.M.M.); sevastita.muste@usamvcluj.ro (S.M.); 2Department of Food Science, Faculty of Food Science and Technology, University of Agricultural Sciences and Veterinary Medicine Cluj-Napoca, 400372 Cluj-Napoca, Romania; carmen-rodica.pop@usamvcluj.ro; 3Department of Polymer Composites, Raluca Ripan Institute for Research in Chemistry, Babeş-Bolyai University, 30 Fântânele Street, 400294 Cluj-Napoca, Romania; miuta.filip@ubbcluj.ro; 4Department of Environmental and Plant Protection, University of Agricultural Sciences and Veterinary Medicine Cluj-Napoca, 400372 Cluj-Napoca, Romania; bogdan.onica@usamvcluj.ro; 5Department of Food Control, Banat’s University of Agricultural Sciences and Veterinary Medicine “King Michael I of Romania”, 300641 Timisoara, Romania; alexa.ersilia@yahoo.ro; 6Department of Chemistry and Biochemistry, Faculty of Animal Science (zootehnie) and Biotechnologie Cluj-Napoca, University of Agricultural Sciences and Veterinary Medicine Cluj-Napoca, 400372 Cluj-Napoca, Romania; gelu.visan@yahoo.com

**Keywords:** proximate composition, micro and macro elements, fatty acids, carbohydrate, Dacia Secuieni, Zenit

## Abstract

Hemp flour from Dacia Secuieni and Zenit varieties was added to bread in different proportions (5%, 10%, 15% and 20%) to improve its nutritional properties. The purpose of this paper was to present the advanced nutritional characteristics of these bread samples. The selected varieties of hemp, accepted for human consumption, met the requirements for the maximum accepted level of THC in seeds. The protein content of new products increased from 8.76 to 11.48%, lipids increased from 0.59 to 5.41%, mineral content from 1.33 to 1.62%, and fiber content from 1.17 to 5.84%. Elasticity and porosity decreased from 95.51 to 80% and 78.65 to 72.24%, respectively. K, Mg, Ca, P, Mn and Fe are the main mineral substances in bread with addition of hemp flour from the Dacia Secuieni and Zenit varieties. The total amount of unsaturated fatty acids in the bread samples with hemp flour ranged from 67.93 g/100 g and 69.82 g/100 g. Eight amino acids were identified, of which three were essential (lysine, phenylalanine, histidine). Lysine, the deficient amino acid in wheat bread, increased from 0.003 to 0.101 g/100 g. Sucrose and fructose decreased with the addition of hemp flour, and glucose has not been identified. The amount of yeasts and molds decreased in the first 3 days of storage. Regarding the textural profile, the best results were obtained for the samples with 5% addition. In conclusion, bread with the addition of hemp flour has been shown to have superior nutritional properties to wheat bread.

## 1. Introduction

Bread is one of the most consumed foods on the planet. Its importance in the development of mankind, in terms of food, is undeniable. It is a staple food product, dating back to Mesopotamia, reminiscent of its consumption during the construction of the great pyramids of Egypt [1,2].

Bread is consumed in various forms around the world, and the average global consumption is 70 kg per year per capita, and in Europe it is 59 kg of bread per year per capita [3].

The success behind the high consumption of bread is given by the simplicity in terms of accessibility of ingredients, preparation, and low cost, and last but not least the multitude of cereals that can be used to bake them [4]. 

Given the growing interest in “redefined” healthy foods, the enrichment of bread with various bioactive compounds, micro- and macroelements is becoming more widespread [5,6]. Among the existent products we mention bread with mushrooms [7], bread fortified with cobia (*Ragycentron canadum*) [8], protein fortified bread with cumin (*Cumin cyminum*) and caraway seeds (*Carum carvi*) [9], soybean bread with chickpeas [10], or gluten-free bread with soybean isolate and calcium caseinate [11].

The concept of “enriching” or “fortifying” a food, according to the Codex Alimentarius, is the addition of one or more nutrients, whether or not they are found in that food. Fortification aims to prevent or correct a nutrient deficiency, in relation to the entire population or to certain groups held (Codex Alimentarius Commission, 1987) [12]. Fortification has been practiced in developed countries since 1940, when cereal products were enriched with niacin, riboflavin, and thiamine [13]. In recent years, and in developing countries, fortification programs have progressed rapidly, becoming increasingly attractive [14].

Svetlana Popel states that the most affordable and effective way to provide various nutrients to the population is to fortify daily foods, especially bakery products. Also, these enrichments should not diminish the shelf life, but neither the sensory nor nutritional qualities of the products [15]. Food fortification is increasingly needed because, according to Liu et al., more than 2 billion people worldwide suffer from micronutrient malnutrition [16].

Fortified products are increasingly preferred because consumer needs and preferences have changed greatly in recent years. The contemporary consumer no longer considers bread just an energy source, wants bread to be a product with adequate nutritional and functional value for the daily diet [17].

Hemp, *Cannabis sativa* L., belongs to the Cannabaceae family and is one of the oldest plants originally cultivated in Central Asia, and from the Bronze Age and in Europe [18,19]. For food use, it is cultivated for its seeds, but the whole plant has applicability in the pharmaceutical industry, agriculture, construction, textile industry, or animal feed [20].

Although some studies show that the two components present in hemp with psychotropic activity, delta-9-tetrahydrocannabinol (THC) and non-psychoactive cannabidiol (CBD), are not found in seeds, only in the leaves, stems, and inflorescence of the female plant [21,22,23,24], there are suspicions in this regard. These substances can be found in seeds only through contamination [25]. From this point of view, only certain varieties, legally approved, can be used for consumption. These varieties are not allowed to exceed 0.2% THC of the dry amount, the weight of the leaves and the flowering parts [20,21,26]. In Europe there are about 70 varieties of hemp *C. sativa* L., approved and regulated by the European Union Plant Variety Database, for human consumption [21,27,28]. Hemp seeds are appreciated and considered a good material for fortification for several reasons. They contain between 25 and 35% lipids, of which 70–80% are polyunsaturated fatty acids [29]. Thehemp seeds contain a significant amount of essential fatty acids, respecting the ideal ratio (between 3:1 and 5:1, depending on the variety) between n-6/n-3. Thus, hemp seed oils help the proper functioning of the body and prevent neurodegenerative diseases, cardiovascular and various types of cancer [29,30,31]. Proteins, rich in essential amino acids, range from 20 to 25% [32,33,34,35,36,37,38]. The predominant proteins are edestin, a storage protein albumin (a globular protein). Another advantage of hemp proteins is their high digestibility, which is improved at a heat treatment of 100 °C for 15 or 30 min [39]. The amount of total carbohydrates varies between 20 and 30%, a large part being dietary fiber, predominant being the insoluble ones [34,36]. A great advantage of these fibers is the benefits on the human body, namely improving insulin sensitivity, helping intestinal transit, reducing appetite, and the risk of obesity and diabetes. They are considered a good probiotic, lower total blood cholesterol, and have anti-inflammatory effects on the large intestine [26]. Mineral substances stand out in significant quantities, which fall between 3.7 and 5.9%. Most of the macroelements in hemp seeds are phosphorus (P), potassium (K), magnesium (Mg), sodium (Na), and calcium (Ca), and in the category of trace elements, the majority were iron (Fe), manganese (Mn), zinc (Zn), and copper (Cu) [20,33,36,37]. Concentrations of micronutrients and macronutrients in cannabis sativa plants vary between varieties and plant organs [40,41] and are affected by cultivation conditions [42,43,44,45].

As the awareness of the nutritional value of hemp seeds began to increase, there was a growing interest in their food use. Thus, there have been more and more recommendations for the introduction of hemp seeds and hemp seed foods in the diet. Although there are several studies on the analysis of hemp seeds in various foods such as bread, biscuits, cookies [21,46,47,48], rice flour bars [49], pork loaves [50], or yogurt [51], the studies are focused on a specific directive, without specific characteristics depending on the variety of hemp used. Thus, the present study aims to make a detailed characterization of the nutritional characteristics of bread with various additions of hemp seed flour, belonging to the varieties Dacia Secuieni and Zenit.

## 2. Results and Discussion

### 2.1. Physico-Chemical Analyses

The results presented in Table 1 illustrate the physico-chemical characteristics for the nine bread samples. Eight of the samples represent the prototypes of functional bread with the addition of 5%, 10%, 15%, and 20% hemp flour from the two varieties studied: Dacia Secuieni and Zenit. In order to highlight the advantages and/or disadvantages of fortifying the classic bread with hemp flour, a control sample (PM) was also included in the study. This control sample was made exclusively from wheat flour, following the same protocol in the manufacturing process. It is considered a control sample. The comparative analysis of the samples was performed in order to explain the impact that the different proportions of hemp flour added in wheat flour may have on the results of the variation.

The bread samples studied had a moisture between 35.98% and 25.63%. As the amount of hemp flour in the bread increase, a decrease in the moisture content was observed. This decrease is due to the low humidity of the hemp flour used 4.49% for hemp flour variety Dacia Secuieni, respectively 4.98% for hemp flour variety Zenit [29]. The same decreasing trend was reported by Mikulec et al. [52] for hemp bread, but they started from a higher humidity of the control sample (45.36%).

The quality of proteins from vegetal sources is an increasingly promoted topic in the field of nutrition. Due to the importance of consuming essential amino acids in a balanced diet, nutritionists recommend the use of vegetable protein sources. Due to the low protein content of wheat flour [17], the addition of hemp flour is auspicious for strengthening this type of food. Bread with the addition of hemp flour comes with a significant intake of high-quality protein, showing increases from 8.76% in the case of the control sample, to 10.48%, and 11.48 for bread with 20% addition of hemp flour. The reported differences were due to the variety from which the hemp seeds come. Slightly higher results, for bread with 15% addition of hemp flour Mikulec et al. [52] reports a percentage of 13.38%, and in our case the samples reach up to 10.7% and 10.45%. However, it should be noted that the control sample in their case had a percentage of 11.02%, with an increase of 3.36% to the addition of 15% hemp flour, and in our case the increase in the percentage of protein for the samples with 15% addition was 1.69%, respectively 1.94%. Similar results were reported by Bădărău et al. [53], with an increase of 1.48% in the case of bread with the addition of 15%. Increases in protein content was reported for bread with mushrooms [7], bread with chickpeas [10], bread with lentils [54] or pasta with mushroom powder [55].

In addition to a significant increase in protein content, bread with the addition of hemp flour also came with a significant lipid intake. There wasan increase from 0.59% in the case of the control sample to 5.31%, respectively 5.41% for the samples with the addition of 20%. A tendency to increase the lipid content wasreported by several authors, but the differences are quite large and are justified by the variety of hemp seeds chosen. Apostol et al. report increase of up to 3.11% in the case of bread with the addition of 20% hemp flour and 2.02% in the case of samples with 10% [56]. For samples with 10% addition, Lukin and Bitiutskikh Ksenia, report a lipid content of 3.61%, and Badarau et al. a percentage of 3.09%, with increases of 1.5–2.0% compared to the control sample, which denotes that the results reported by us are consistent with those reported in the literature [57]. However, Pojic et al. report lower values and increases of only 0.4% for bread with the addition of 10% hemp flour [58].

The ash content reported in the bread samples with hemp flour shows a slight increase of 0.27% for hemp flour variety Dacia Secuieni, respectively 0.29% for hemp flour variety Zenit, compared to the control sample. This increase is due to the relatively high content of mineral substances in hemp flour. Apostol et al. [56], report higher increase than those reported in this study, up to 1.34% in the case of samples with 20% hemp flour, compared to the control sample.

Similar to the results obtained in the case of proteins and lipids, the fiber content shows an increase from 1.17% in the case of the control sample, up to 5.39%, respectively 5.84%. In the case of samples who 10% addition of hemp flour, an increase of 2.37% is observed, respectively 2.12% compared to the control sample. Our results are lower than those reported by Lukin et al., who reported increase of 3.48% or 4.31% reported by Apostol et al., reporting the same percentage of the addition. In the case of total fibers, higher results were reported than those obtained for the Dacia Secuini and Zenit varieties [56,57]. These differences are due to the variety from which hemp seeds come, but most studies do not show results specific to the varieties of which hemp seeds are part.

In the case of acidity, there wasa slight increase in the case of samples with added flour from the Dacia Secuinei variety and a higher increase in the case of samples with added flour from the Zenit variety, compared to those mentioned above. The increasewith a maximum of 0.4% in the case of samples coming from the Dacia Secuieni variety, and a maximum of 0.9% in the case of those coming from the Zenit variety. Similar results were reported by Bădărău et al. [53] and Apostol et al. [56], with increase of 0.55% and 0.4% for bread with the addition of 15% hemp flour, respectively. The salt content shows small, statistically insignificant fluctuations.

Another advantage of adding hemp flour to bread is the increase in energy value from 253.71 kcal/100 g to 311.95 kcal/100 g, respectively, 316.81 kcal/100 g, the date of high-quality nutrients. The same trend is reported in the case of fortifying bread with mushroom powder [7].

The results presented in Table 2 on Pearson correlations show a positive significance for PM with humidity. In the case of raw proteins, a positive correlation was identified with the increase of the added content of hemp flour, regardless of variety. The same positive correlation was observed in the case of lipids, ash, fiber, acidity, NaCl, and energy value, for the samples with 20% addition of hemp flour.

### 2.2. Porosity and Elasticity of Bread Samples

Hemp flour is rich in fiber and protein, and its addition to the mix with wheat flour has a significant influence on rheological parameters, such as elasticity and porosity, reflected in the quality of the finished product. The addition of hemp flour to bakery products had as a side effect the decrease of their elasticity and porosity. The influence of hemp flour on the porosity and elasticity of the bread is presented in Table 3. In both cases it indicated that there is an inversely proportional relationship between the two variables. Thus, both the elasticity and the porosity of the samples decreased with the increase of the ratio of added hemp flour. These decreases were due to the absence of gluten in hemp flour. Apostol et al. reported decreases in porosity and elasticity [56]. A minor decrease in porosity has been reported in other types of bread enriched with pumpkin [59] or corn germ flour [60]. A decrease in elasticity from 94.82% to 89.04% has been reported for bread enriched with mushroom flour [7].

Table 4 presents pearson correlations between porosity and elasticity. In the case of porosity, there were positive correlations for the samples with flour from the Zenit variety and negative correlations, which increased for the samples with more than 10% of hemp flour added from the Dacia Secuieni variety. In the case of elasticity, the negative correlations were noticed in the case of samples with 15 and 20% addition of hemp flour, regardless of variety.

### 2.3. Content of Micro and Macro Elements

The mineral elements determined in the bread samples were Cu, Cd, Cr, Ni, Pb, Zn, Fe, Mn, Ca, Mg, K, and P. Of these, in all samples a significantly higher Ca content was identified and of K. The bread obtained by adding 20% hemp flour from the Dacia Secuinei variety had the highest percentage in Ca, and the one with the addition of 20% hemp flour Zenit variety had the highest percentage in the case of K. However, it was the quantity of Mg which reached up to 394.33 mg/100 g. The smallest amounts of micro and macroelements were recorded in the case of Cu, Cd, Cr, Ni, Pb, Zn, Fe, Mn (Table 5).

Enrichment with micro and macro elements of bread samples with the addition of hemp flour was also reported in the literature. Lukin et al. [57] reported a lower content of Ca and Mg, and a higher amount of P, from the 4 elements studied. Apostol et al. [56] identified a higher amount of Mg and less of Ca and K, and Bădărău et al.’s [53] research showed higher amounts of K and Mg in the bread samples with hemp flour studied.

Table 6 showed the Pearson correlations for the content of micro and macroelements. There was a very strong correlation for the 20% PFCZ sample with the variables Cd and Ni. In the case of the control sample, the results obtained showed that it significantly negatively influences the mineral elements compared to the bread variants with the addition of hemp flour.

### 2.4. Fatty Acids Content

Table 7 shows the fatty acids in the bread samples with hemp flour in different proportions and the bread control sample. Nine saturated and unsaturated fatty acids were identified. The saturated fatty acids were palmitic acid, stearic acid, myristic acid, margaric acid, and arachidic acid. Among the identified unsaturated fatty acids, we mention linoleic acid, oleic acid, gamma-linolenic acid, and palmitoleic acid.

The highest values were given by linoleic acid. It can be seen that it a decrease can be observed with the addition of hemp flour. Most studies performed on hemp seeds or flours obtained from hemp seeds reported the presence of an unsaturated fatty acid content of up to 90% [21,29,61]. It is known that the consumption of fatty acids is closely related to the occurrence of coronary heart disease, metabolic syndromes and changes in the intestinal microbiota, while the consumption of unsaturated fatty acids has health-improving effects [62,63,64].

Among the unsaturated fatty acids, linoleic acid was followed by oleic acid, gamma-linoleic acid, which was constantly increasing with the addition of hemp flour, and palmitoleic acid. The total amount of unsaturated fatty acids in the bread samples with hemp flour were between 67.93 g/100 g and 69.82 g/100 g. They can be monounsaturated (MUFA) and polyunsaturated (PUFA), the polyunsaturated ones being found in proportion of about 50%. Linoleic acid (18:2, n-6, LA) and gamma-linoleic acid (18:3, n-6, GLA), the main PUFA acids, belong to the category of essential fatty acids (EFA). These acids are essential because they cannot be synthesized by the human body and must be assimilated from the diet, they are indispensable for the normal functioning of the human body [21,29,30]. Consequently, bread with the addition of hemp seed flour was considered an adequate source of EFA.

In terms of the amount of saturated fatty acids, they were found in quantities of less than 30% in fortified bread samples and were constantly decreasing with the addition of hemp seed flour.

The results on the amounts of saturated and unsaturated fatty acids were reported by Apostol et al., who reported a higher percentage than reported by us of unsaturated fatty acids of up to 84.77% in the case of bread with the addition of hemp flour of 20% [56]. It should be mentioned in this study that the control sample had 81.25% unsaturated fatty acids.

According to the data in Table 8, there was a very significant correlation in the sample control that positively influences the values of myristic, palmitic, oleic, arachidic, and palmitoleic acid, but having significantly negative influences on the other three fatty acids. In the case of bread variants with the addition of Dacia Secuieni hemp flour we saw that for the sample with 5% significant positive influences are found in the case of linoleic acid, but negative for five other fatty acids, and for the sample with 20% positive influences are found in the case of oleic and gamma-linolenic acids. Regarding the samples with the addition of flour from the Zenit variety, positive correlations were observed in the samples with 5 and 10% addition, for myristic acid. Significant positive correlations were observed in the case of samples with 20% addition for stearic, gamma-linoleic, and palmitoleic acid.

### 2.5. Amino Acids Content

The quality of proteins was given by the composition of amino acids and their availability. Due to the promotion by nutritionists of the quality of vegetable proteins, plant fortified products are in great demand. Due to the importance of essential amino acids in the body, many nutritionists recommend the consumption of proteins of vegetable origin. The low content of wheat protein, especially in essential amino acids such as lysine, makes wheat flour protein considered inferior [17].

A total of 8 amino acids were identified in the bread samples, of which 3 are essential (lysine, phenylalanine, histidine) and 5 non-essential (arginine, glutamic acid, cysteine, tyrosine, aspartic acid), according to Table 9. Given the lysine deficiency from bread and the statistically significant increase of this essential amino acid together with the addition of hemp flour from 0.003 g/100 g to 0.105 g/100 g, we can observe the importance of fortifying bread with hemp seed flour. In addition to the increase in lysine content, there was a proportional increase in the other 2 essential amino acids with the addition of hemp flour.

In the case of non-essential amino acids, we observed a constant increase for all, directly proportional to the addition of hemp flour. The most significant increase being reported for arginine and glutamic acid.

The above results are justified by the high amounts of amino acids present in the flours obtained from the seeds of the hemp varieties Dacia Secuinei and Zenit [29].

Regarding the Pearson correlations for amino acids of bread samples presented in Table 10, they reinforce those stated in Table 9, namely that the control bread had significantly negative correlations in all samples. Positive correlations were noticeable with the addition of hemp flour, regardless of variety. The most representative correlations were noted in the case: PFCDS 15% (for phenylalanine), PFCDS 20% (for histidine, glutamic acid, cysteine and tyrosine), and PFCZ 20% (for lysine histidine or cysteine.

### 2.6. Carbohydrate Content

Sucrose and fructose were quantified in the bread samples. Glucose was not found. It was observed according to Table 11 that a significantly high content in all bread samples refers to sucrose. The fructose content was very low both in the control sample and in the samples of bread fortified with hemp flour and wheat flour. Compared to the results obtained in the case of hemp and wheat flours, both the sucrose and the fructose content of the bread samples was lower [29]. The main cause was the process of baking bread where carbohydrates are converted due to the Maillard reaction, in conditions of low humidity and high temperature.

Decreased concentrations of carbohydrates (glucose, fructose, sucrose), fermentable carbohydrates, were constantly changing throughout the technological process, both due to the amylolytic process and the consumption of yeast during the process of fermentation and bread production [65].

Glucose was not detected in the wheat bread or hemp flour fortified samples. Similar results were reported by Carocho et al. [3] for wholemeal bread and oat bread and detected in very small quantities in multigrain bread, Bavaria or rice bread. This could be due to the low glucose level present in most cereals [66] and sustained by the long fermentation time, thus consuming a significant amount of the small amount of glucose available.

Regarding the Pearson correlations for carbohydrate of of bread samples (Table 12) semipositive results were observed in the case of the control sample for both sucrose and fructose, and the data with the addition of hemp seed flour shows negative correlations, regardless of variety, to the addition of 15 and 20%.

### 2.7. Microbiological Analyses

The amount of yeasts and molds in the bread samples had a decreasing tendency with the addition of hemp flour, regardless of variety (Table 13). They had the same path, in some cases statistically insignificant, with the short-term storage.

This decrease during storage was due to the low moisture content of the bread samples (Table 1). This conclusion was also supported by Pearson’s correlation between humidity and microbiological content, where according to Table 14, all correlations are positive.

The inhibition of yeasts, molds and the development of gram-positive and negative bacteria was also due to the presence of lactic acid in bread or other fermented products [67,68].

Regarding the Pearson correlations for the microbiological parameters) of bread samples (Table 15), there were positive correlations for PM, PFCDS 5%, and PFCDS 10%, and for the rest of the samples the correlations obtained were negative.

### 2.8. Texture Analysis of Bread Samples

One of the main selection parameters of the quality of a food product is given by its texture, which is the analysis from different points of view such as sensory, total appearance, shelf life, or nutritional value [69].

According to Table 16 on the texturometric result, 9 parameters from the bread samples were analyzed, namely, cycle duration 1, total duty cycle 1, cycle duration 2, cohesiveness, total duty cycle 2, spontaneity index, viscosity and chewability.

Through these analyses, the hardness, cohesiveness, viscosity, and chewability of the bread with the addition of hemp flour in different proportions from the Dacia Secuieni and Zenit varieties were evaluated. Firmness was accepted as a measure of the freshness and quality of bread, and hardness is the main mechanical characteristic for the consumer when consuming solid foods, being defined as the force needed to achieve a certain deformation of the finished product [70].

The apparent increase in the hardness of cycle 1, but also of cycle 2 was due either to the reduction of the amount of gluten or to the emulsifying property of hemp flour which helps to improve the hardness parameter, even if the addition of hemp flour reaches 20%. The Zenit 20% hemp flour bread sample was the one that behaves best in terms of hardness, recording the lowest value of 387.

A possible explanation for the decrease in hardness as the hemp flour content increase is its oil content, an oil that can improve the texture of the finished product [58]. Cohesion is defined as the parameter that reflects the internal cohesion of the finished product. The higher the value of cohesiveness, the more the product does not disintegrate during the mastication process [65].

The degree of cohesiveness around 0.67 is due to the addition of water. Both the control sample bread and the PFCDS 10% recorded the same value 0.69 being the highest value recorded. Regarding the viscosity parameter, it can be observed that the highest percentage is presented by P PFCDS 5% (612.67) compared to PFCDS 10% with the lowest percentage (221.67).

Chewability is defined as the energy required to perform the chewing process until the beginning of the swallowing process [71]. The sample that recorded the highest value for the chewability index was PFCDS 5%, and the lowest percentage was PFCZ 20%.

The influence of the addition of hemp flour in bread on hardness has also been reported by Švec and Hrušková [70] and Korus et al. [46,47], Anna Mikulec et al. [56]. Fluctuations of the whole date texture with different additions were also reported, for example, fenugreek bread [72] or lentils [73].

Table 17 presented Pearson correlation for texture profile analyzes content of bread samples. In the case of hardness cycle 2, total work cycle 2, viscosity and chewiness, a very strong correlation was remarked for PFCDS 5%, but also a springiness index for PFCZ 10%.

## 3. Materials and Methods

### 3.1. Materials

All ingredients used to make the bread were purchased from the local market in Cluj Napoca, Romania. The flour usedwas type 550 wheat. The type of flour depends on the ash content, according to the Romanian Standardization Association [1]. Hulled hemp seeds (*Cannabis Sativa* L.) from which hemp flour (obtained by grinding) was obtained belong to the Dacia Secuieni and Zenit varieties. They were purchased from the Secuieni Agricultural Research–Development Station, Neamț. These two varieties of hemp (*Cannabis Sativa* L.) are among the 70 varieties accepted in Europe. They were purchased with certificates of analysis stating that these seeds have a delta-9-tetrahydrocannabinol (THC) content <0.2%. The advanced characterization of these varieties is presented by Rusu et al. [29].

The first step was to make flour mixes according to Table 18, in which wheat flour was substituted with hemp flour in different percentages, of the two varieties. The technological parameters are presented in Table 18 and the final products are in accordance with Figure 1.

### 3.2. Proximate Composition

Proximate composition (moisture, protein, ash, crumb porosity and elasticity) were determined according to Romanian official methods SR 91: 2007 [74]. The nitrogen (N) content was determined using the Kjeldhal apparatus, and the conversion factor for the calculation of the crude protein was 5.7 according to SR ISO 1871/2002 [75] for the protein in vegetable products. The determination of the lipid content was performed according to SR ISO 6492: 2001 [76]. According to STAS 90/1988 [77], Romanian official methods, acidity was determined by the 67% (*v*/*v*) ethyl alcohol method and the crude fiber content was determined according to ISO 5498: 1981 [78]. The energy value was calculated according to the formulas described by Vlaic et al. [7].

### 3.3. Determination of Micro and Macroelements

Determination of the content of micro and macroelements by atomic absorption spectrometry were determined according to SR EN 14082:2003 [79,80].

### 3.4. Determination of Fatty Acid Composition

An application used was the GC—MS system (gas—chromatograph coupled with mass spectrophotometer) of a sensitive analytical method to be able to identify fatty acids from composite flour samples. With the help of the BF3-MeOH method, the methyl esterification of the samples used in the analysis was performed, thattakes place after the alkaline hydrolysis process. For a 20 mg oil sample 2 mL 0.5 mol/L methanol solution was added, the mixture being heated for 7 min at 100 °C. After cooling, 3 mL of BF3-MeOH having a concentration of 14% is added, and the vessel is heated for 5 min at 100 °C. After cooling, 7 mL of saturated NaCl solution is added, after which the solution is extracted. The resulting hexane layer (2 mL) was used as a solution for gas chromatography [81]. The FAME analysis was performed based on GC-MS QP 2010 using the Shimadzu system. It was equipped with an automatic injector with and without splitting the mobile phase flow split/splitless. Separation was performed using a Zebron ZB-FFAP capillary column (60 m × 0.25 mm ID, 0.25 μm film thickness). Helium was used as a carrier gas with a flow rate of 1.99 mL/min, with a split ratio of 1:10. The injector temperature reached was 250 °C. The oven temperature was programmed for 10 min at 140°C, after which it was raised to 250 °C with a rate of 7 °C/minute, after which the final temperature was maintained for 10 min. The control of the GC—MS operation was possible thanks to the software from LabSolution. MS spectra were assimilated in a range having a width of 40–500 *m*/*z*, the ion source temperature was 210 °C the interface temperature was 255 °C, the solvent was interrupted in 3 min, the scanning speed was 2500 units/s, and the actuation time was 0.2 s. The peaks of the fatty acid methyl esters were determined by comparing the equivalent chain length with the retention time. FAME standards from Supelco Inc., Bellefonte, PA, USA (Supelco 37 component FAME Mix) and other reagents from Merck, Germany were used. A total of 3 determinations were performed.

### 3.5. Determination of Amino Acids

Determination of amino acids using high performance liquid chromatography (HPLC) involves their acid hydrolysis in the presence of 6 M HCL except for sulfur amino acids, identification and chromatographic dosing where the DIONEX ICS—3000 amino acid analyzer is used. Take 0.5 g of the sample and hydrolyze with 10 mL of 6 M HCl at 100 °C for 24 h. The sample is filtered through a 0.2 μm Milipore filter and the sample is diluted in a ratio of 1:10 or 1:1000 with 0.1 N HCl depending on the nature of the sample, finally injected into the chromatograph. Chromatographic conditions: chromatographic column Pre-column AMINOPAC PA10 (2 × 50 mm, P/N 055407). Mobile phases: E1: water; E2: 250 mM NaOH and E3: 1 M NaAc, gradient, Mobile phase flow (0.25 mL/min.; Reference electrode: pH/Ag/AbCl, column temperature being 30 °C.

### 3.6. Determination of Individual Carbohydrate

The HPLC-RI method consists in separating at a temperature of 70 °C on a column of CARBOSep COREGel 87C (300 × 7.8 mm) with CARBOSep 87C cartridge and CARBOSep COREGel 87C cartridge. The mobile phase was represented by ultrapure Milipore water, the flow rate was 0.5 mL/min., For the determination of glucose, sucrose, and fructose, the injector volume was 20 µL. The process of extracting carbohydrates from the samples studied was performed in water. Three g of the sample was weighed and grinded well, then 8 mL of extraction solvent water was added. The mixture was ultrasonified for 15 min and then centrifuged for a period of 20 min at 4500 rpm. The supernatant was filtered through a 0.45 μm filter and injected into HPLC. Carbohydrate amounts were expressed in mg carbohydrate/100 g sample. The following materials were used in the determination of carbohydrates: acetonitrile (HPLC purity), purchased from Merck (Darmstadt, Germany); the standards of resveratrol, fructose, sucrose, and glucose were purchased from the United States, from Milwaukee, Aldrich, Milipore ultrapure water (18.2 MΩcm). The equipment used consisted of Jasco HPLC chromatograph (Japan) for analysis, it is equipped with an HPLC pump (Model PU-980), a column thermostat Model CO-2060 Plus, ternary gradient unit Model LG-980-02, UV-VIS detector (Model UV-975), injection valve equipped with a 20 µL test loop (Rheodyne) and detector with refractive index Model RI-2031 Plus. The samples were injected with a Hamilton Rheodyne syringe (50 mL). The HPLC system was controlled, and the experimental data was performed using ChromPass software. For the construction of the calibration curve a standard mixture of carbohydrates with different dilutions (5 concentrations) was used, with a quality between 333 µg/ml and 2000 µg/mL. This HPLC-RI method developed by Filip et al. [82], was improved and optimized to identify and quantify the carbohydrates in flours samples.

### 3.7. Determination of the Total Number of Yeasts and Molds

ISO 21527-2:2008 standard method was used [83]. An amount of 5 g sample was weighed into a sterile stomacher bag using sterile instruments. Forty-five mL of peptone saline diluent (Oxoid Ltd., Basingstoke, Hampshire, England) was added and homogenized in a stomacher (Bag Mixer 100 MiniMix, Interscience, St. Nom, France) for 1 min to prepare a 10–1 homogenate. Further decimal dilution (10–2) was prepared as well in peptone saline diluent. All dilutions were inoculated in duplicate.

Briefly, 0.1 mL of the diluted sample (10−1 and 10−2) was transferred into a sterile Petri dish covered with Dichloran Glycerol Agar (Oxoid, Basingstoke, UK) and spread using a Drigalsky spatula. The plates were incubated for 5 days h at 25°C. Typical colonies of yeasts and molds were counted after 5 days using a colony counter (Colony Star 8500, Funke Gerber, Berlin, Germany).

### 3.8. Texture Analysis

The main objective of the method was to measure the firmness of sliced bread as a result of its storage (freshness vs. aging). The test was based on the theory that an increase in load and the decrease in compressibility occurs with the storage of bread. This adaptation of the method of compressing the bread samples not only provides valuable information on the storage conditions of the product but is also a valuable indication of the differences in texture caused by the production process and the recipe. A slice of bread 25 mm thick or two slices 12.5 mm thick are placed under a cylindrical probe with a diameter of 38.1 mm. The bread is compressed 3 mm, and the tip of the texture curve is used as an indication of freshness. The sample can also be compressed to a target load. The distance traveled to reach the load is defined bycompressibility (the ability to compress), being a recommendation of freshness. The bread sample is placed centrally, horizontally on the table of the texture analyzer. During the analysis, the treatment performed on the compression load provided data that was continuously recorded on the computer using software.Texture analyzer settings: target distance: 3.0 mm; target standby time: 0 s; minimum detectable load: 4 g; probe speed during test: 1.00 mm/s; probe speed after test: 4.5 mm/s; number of test cycles: 1.0; probe speed before the start of the test: 0.5 mm/s; device used: Brookfield CT3 Texture Analyzer; probe used: cylinder TA4/1000 with the following characteristics: 38.1 mm diameter cylinder probe, clear acrylic 26 g. 20 mm long.

### 3.9. Statistical Analysis

Statistical data analysis was performed using statistical interpretation programs R Studio version 3 [84] and StatSoft Statistics version 12 (STATSOFT, 2002). With the help of the StatSoft Statistics program version 12, Pearson correlations were made, and through the R Study program version 3, the Fisher LSD test was performed using the “agricolae” package [85].

Pearson correlations. The correlation in the broadest sense is defined as a measure of an association between variables. Changing the size of one variable in correlated data is associated with changing the size of another variable, either in the same direction (positive correlation) or in the opposite direction (negative correlation). The term correlation is used in the context of a linear relationship between two continuous variables and expressed as a Pearson product-moment correlation [86].

## 4. Conclusions

Given the growing interest in fortified products and at the same time for the use of hemp seeds in human consumption, this study showed that the addition of flour from hemp seeds in bread improves its nutritional properties. It is very important to note that these properties can be influenced by the hemp variety. This bread fortification process has led to a high content of proteins and essential amino acids, lipids and unsaturated fatty acids, fiber, and minerals, not significantly affecting the rheological properties of the finished product. The results of this study showed the significance the use of hemp seed flour in a product used daily. So, when hemp seed flour was used as a source of nutrients in fortified bread making, it added value in diversifying the assortment range and obtaining new nutritional assortments.

## Figures and Tables

**Figure 1 plants-10-01558-f001:**
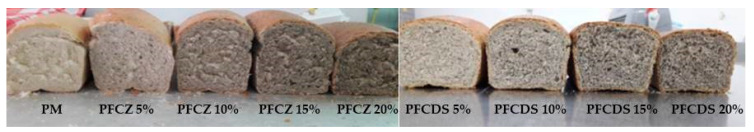
Final products. * PM—control sample; PFCDS 5%—bread with 5% hemp flour Dacia Secuieni; PFCDS 10%—bread with 10% hemp flour Dacia Secuieni; PFCDS 15%—bread with 15% hemp flour Dacia Secuieni; PFCDS 20%—bread with 20% hemp flour Dacia Secuieni; PFCZ 5%—bread with 5% hemp flour Zenit; PFCZ 10%—bread with 10% hemp flour Zenit; PFCZ 15%—bread with 15% hemp flour Zenit; PFCZ 20%—bread with 20% hemp flour Zenit.

**Table 1 plants-10-01558-t001:** Physico-chemical properties of bread samples.

Samples	Moisture (%)	Crude Protein (%)	Lipids (%)	Ash (%)	Crude Fiber (%)	Acidity (°acidity)	NaCl (%)	Total Carbohydrate [g/100 g]	Energy Value [kcal/ 100 g]
PM	35.98 ± 0.19 ^a^	8.76 ± 0.07 ^d^	0.59 ± 0.03 ^c^	1.33 ± 0.07 ^c^	1.17 ± 0.06 ^e^	1.3 ± 0.20 ^d^	1.34 ± 0.01 ^abc^	53.34 ± 0.39 ^d^	253.71 ± 0.40 ^e^
PFCDS 5%	30.72 ± 0.07 ^bc^	10.31 ± 0.06 ^abc^	0.95 ± 0.05 ^bc^	1.35 ± 0.04 ^c^	2.03 ± 0.08 ^de^	1.5 ± 0.10 ^bcd^	1.33 ± 0.04 ^bc^	56.67 ± 0.93 ^bc^	276.43 ± 0.22 ^d^
PFCDS 10%	27.74 ± 0.08 ^c^	10.49 ± 0.08 ^abc^	1.66 ± 0.04 ^bc^	1.40 ± 0.08 ^c^	3.54 ± 0.09 ^bcd^	1.7 ± 0.20 ^abcd^	1.35 ± 0.04 ^abc^	58.69 ± 0.84 ^ab^	291.54 ± 0.30 ^cd^
PFCDS 15%	27.43 ± 0.10 ^c^	10.70 ± 0.12 ^ab^	2.12 ± 0.05 ^bc^	1.55 ± 0.09 ^ab^	4.44 ± 0.08 ^abc^	1.9 ± 0.10 ^ab^	1.37 ± 0.05 ^ab^	58.19 ± 0.12 ^ab^	294.44 ± 0.26 ^bcd^
PFCDS 20%	26.75 ± 0.09 ^d^	11.48 ± 0.10 ^a^	5.31 ± 0.06 ^a^	1.60 ± 0.05 ^a^	5.39 ± 0.08 ^ab^	2 ± 0.20 ^a^	1.39 ± 0.04 ^a^	54.86 ± 0.74 ^cd^	311.95 ± 0.27 ^ab^
PFCZ 5%	28.52 ± 0.07 ^bc^	9.44 ± 0.12 ^cd^	0.53 ± 0.04 ^c^	1.37 ± 0.05 ^c^	2.43 ± 0.07 ^de^	1.4 ± 0.20 ^cd^	1.31 ± 0.05 ^c^	60.14 ± 0.22 ^a^	283.09 ± 0.15 ^d^
PFCZ 10%	27.11 ± 0.04 ^c^	9.96 ± 0.04 ^bcd^	2.1 ± 0.18 ^bc^	1.43 ± 0.06 ^bc^	3.29 ± 0.07 ^cd^	1.6 ± 0.10 ^abcd^	1.33 ± 0.02 ^bc^	59.40 ± 0.24 ^ab^	296.14 ± 0.60 ^bcd^
PFCZ 15%	26.15 ± 0.04 ^c^	10.45 ± 0.03 ^abc^	3.18 ± 0.05 ^ab^	1.58 ± 0.08 ^a^	4.57 ± 0.11 ^abc^	1.8 ± 0.10 ^abc^	1.35 ± 0.03 ^abc^	58.64 ± 0.44 ^ab^	304.58 ± 0.42 ^abc^
PFCZ 20%	25.63 ± 0.09 ^d^	10.48 ± 0.04 ^abc^	5.41 ± 0.04 ^a^	1.62 ± 0.04 ^a^	5.84 ± 0.16 ^a^	1.9 ± 0.20 ^ab^	1.38 ± 0.04 ^ab^	56.86 ± 0.62 ^abc^	316.81 ± 0.37 ^a^

Identical superscripts letters within rows indicate no significant difference (*p* > 0.05); * PM—control sample; PFCDS 5%—bread with 5% hemp flour Dacia Secuieni; PFCDS 10%—bread with 10% hemp flour Dacia Secuieni; PFCDS 15%—bread with 15% hemp flour Dacia Secuieni; PFCDS 20%—bread with 20% hemp flour Dacia Secuieni; PFCZ 5%—bread with 5% hemp flour Zenit; PFCZ 10%—bread with 10% hemp flour Zenit; PFCZ 15%—bread with 15% hemp flour Zenit; PFCZ 20%—bread with 20% hemp flour Zenit.

**Table 2 plants-10-01558-t002:** Pearson correlation for physico-chemical properties of bread samples.

Samples	Moisture	Crude Protein	Lipids	Ash	Crude Fiber	Acidity	NaCl	Total Carbohydrate	Energy Value
PM	0.769	−0.710	−0.370	−0.451	−0.589	−0.503	−0.120	−0.675	−0.737
PFCDS 5%	0.150	0.039	−0.298	−0.387	−0.383	−0.237	−0.240	−0.124	−0.296
PFCDS 10%	−0.198	0.126	−0.155	−0.226	−0.022	0.030	0.000	0.210	−0.003
PFCDS 15%	−0.236	0.227	−0.062	0.258	0.193	0.296	0.240	0.127	0.053
PFCDS 20%	−0.120	0.604	0.581	0.419	0.420	0.429	0.480	−0.424	0.393
PFCZ 5%	−0.109	−0.381	−0.382	−0.322	−0.288	−0.370	−0.480	0.450	−0.167
PFCZ 10%	−0.275	−0.130	−0.066	−0.129	−0.082	−0.103	−0.240	0.328	0.021
PFCZ 15%	−0.387	0.106	0.152	0.354	0.224	0.163	0.000	0.202	0.250
PFCZ 20%	0.406	0.121	0.601	0.483	0.527	0.296	0.360	−0.093	0.487

* PM—control sample; PFCDS 5%—bread with 5% hemp flour Dacia Secuieni; PFCDS 10%—bread with 10% hemp flour Dacia Secuieni; PFCDS 15%—bread with 15% hemp flour Dacia Secuieni; PFCDS 20%—bread with 20% hemp flour Dacia Secuieni; PFCZ 5%—bread with 5% hemp flour Zenit; PFCZ 10%—bread with 10% hemp flour Zenit; PFCZ 15%—bread with 15% hemp flour Zenit; PFCZ 20%—bread with 20% hemp flour Zenit.

**Table 3 plants-10-01558-t003:** Porosity and elasticity (%) of bread samples.

Samples	Porosity (%)	Elasticity (%)
PM	78.65 ± 0.05 ^a^	95.51 ± 0.02 ^a^
PFCDS 5%	76.88 ± 0.04 ^abc^	95.58 ± 0.04 ^a^
PFCDS 10%	74.78 ± 0.05 ^bcd^	93.54 ± 0.01 ^ab^
PFCDS 15%	74.45 ± 0.07 ^cd^	86.44 ± 0.03 ^bcd^
PFCDS 20%	72.24 ± 0.01 ^d^	80 ± 2.00 ^d^
PFCZ 5%	78.60 ± 0.02 ^a^	93.54 ± 0.02 ^ab^
PFCZ 10%	78.26 ± 0.03 ^ab^	91.37 ± 0.04 ^abc^
PFCZ 15%	78.05 ± 0.03 ^abc^	89.09 ± 0.04 ^abc^
PFCZ 20%	77.97 ± 0.06 ^abc^	85 ± 4.00 ^cd^

Identical superscripts letters within rows indicate no significant difference (*p* > 0.05); * PM—control sample; PFCDS 5%—bread with 5% hemp flour Dacia Secuieni; PFCDS 10%—bread with 10% hemp flour Dacia Secuieni; PFCDS 15%—bread with 15% hemp flour Dacia Secuieni; PFCDS 20%—bread with 20% hemp flour Dacia Secuieni; PFCZ 5%—bread with 5% hemp flour Zenit; PFCZ 10%—bread with 10% hemp flour Zenit; PFCZ 15%—bread with 15% hemp flour Zenit; PFCZ 20%—bread with 20% hemp flour Zenit.

**Table 4 plants-10-01558-t004:** Pearson correlation for porosity and elasticity of bread samples.

Samples	Porosity	Elasticity
PM	0.327	0.376
PFCDS 5%	0.036	0.381
PFCDS 10%	−0.307	0.242
PFCDS 15%	−0.361	−0.244
PFCDS 20%	−0.723	−0.685
PFCZ 5%	0.319	0.242
PFCZ 10%	0.263	0.093
PFCZ 15%	0.229	−0.063
PFCZ 20%	0.216	−0.343

* PM—control sample; PFCDS 5%—bread with 5% hemp flour Dacia Secuieni; PFCDS 10%—bread with 10% hemp flour Dacia Secuieni; PFCDS 15%—bread with 15% hemp flour Dacia Secuieni; PFCDS 20%—bread with 20% hemp flour Dacia Secuieni; PFCZ 5%—bread with 5% hemp flour Zenit; PFCZ 10%—bread with 10% hemp flour Zenit; PFCZ 15%—bread with 15% hemp flour Zenit; PFCZ 20%—bread with 20% hemp flour Zenit.

**Table 5 plants-10-01558-t005:** The content of micro and macroelements (mg/100 g) of bread samples.

Samples [mg/100 g]	Cu	Cd	Cr	Ni	Pb	Zn	Fe	Mn	Ca	Mg	K	P
PM	0.33 ± 0.02 ^d^	0.01 ± 0 ^a^	0.02 ± 0.01 ^d^	n.d.	0.11 ± 0.04 ^b^	3.03 ± 0.02 ^b^	12.73 ± 0.02 ^b^	5.91 ± 0.03 ^d^	693 ± 2.00 ^c^	134.67 ± 0.02 ^c^	713.43 ± 3.00 ^a^	55.33 ± 0.06 ^d^
PFCDS 5%	0.52 ± 0.02 ^bcd^	n.d.	0.13 ± 0.03 ^bcd^	n.d.	0.4 ± 0.04 ^a^	5.08 ± 0.04 ^a^	21.87 ± 0.03 ^a^	7.2 ± 0.02 ^bcd^	1085.67 ± 0.05 ^b^	236 ± 3.00 ^ab^	710.33 ± 0.02 ^a^	61.67 ± 0.05 ^cd^
PFCDS 10%	0.61 ± 0.04 ^abc^	n.d.	0.23 ± 0.05 ^ab^	n.d.	0.44 ± 0.03 ^a^	5.39 ± 0.04 ^a^	22.64 ± 0.02 ^a^	8.79 ± 0.04 ^bcd^	1286 ± 3.00 ^ab^	265.67 ± 0.08 ^ab^	627.33 ± 4.00 ^ab^	119 ± 2.00 ^abcd^
PFCDS 15%	0.73 ± 0.09 ^ab^	n.d.	0.27 ± 0.05 ^ab^	n.d.	0.49 ± 0.06 ^a^	5.42 ± 0.05 ^a^	23.11 ± 0.04 ^a^	11.62 ± 0.07 ^bcd^	1345.67 ± 0.03 ^ab^	281.2 ± 0.07 ^ab^	624.33 ± 0.04 ^ab^	124.67 ± 0.02 ^abc^
PFCDS 20%	0.79 ± 0.08 ^a^	n.d.	0.3 ± 0.03 ^a^	n.d.	0.54 ± 0.05 ^a^	5.79 ± 0.03 ^a^	23.86 ± 0.01 ^a^	14.96 ± 0.05 ^a^	1394.67 ± 0.02 ^a^	298.33 ± 0.2 ^a^	593 ± 0.02 ^ab^	184.67 ± 0.04 ^a^
PFCZ 5%	0.38 ± 0.06 ^cd^	n.d.	0.07 ± 0.02 ^cd^	n.d.	0.37 ± 0.02 ^a^	4.88 ± 0.04 ^a^	21.57 ± 0.02 ^a^	6.95 ± 0.03 ^cd^	1064.67 ± 0.06 ^b^	234 ± 2.00 ^b^	591.2 ± 0.01 ^ab^	54.33 ± 0.04 ^d^
PFCZ 10%	0.55 ± 0.04 ^abcd^	n.d.	0.12 ± 0.04 ^bcd^	0.21 ± 0.05 ^a^	0.42 ± 0.05 ^a^	5.33 ± 0.21 ^a^	22.61 ± 0.03 ^a^	7.79 ± 0.06 ^bcd^	1278.33 ± 0.02 ^ab^	261.33 ± 0.04 ^ab^	443.33 ± 0.03 ^b^	115.67 ± 0.09 ^bcd^
PFCZ 15%	0.67 ± 0.09 ^ab^	n.d.	0.19 ± 0.04 ^abc^	0.25 ± 0.08 ^a^	0.46 ± 0.05 ^a^	5.34 ± 0.02 ^a^	22.97 ± 0.03 ^a^	10.55 ± 0.03 ^abc^	1342.67 ± 0.03 ^ab^	278.33 ± 0.07 ^ab^	435.67 ± 0.06 ^b^	121.33 ± 0.02 ^abcd^
PFCZ 20%	0.74 ± 0.05 ^ab^	0.01 ± 0.01 ^a^	0.24 ± 0.05 ^ab^	0.29 ± 0.05 ^a^	0.51 ± 0.09 ^a^	5.68 ± 0.03 ^a^	23.53 ± 0.02 ^a^	13.71 ± 0.04 ^a^	1391.33 ± 0.05 ^a^	293 ± 5.00 ^ab^	180 ± 0.02 ^c^	182.34 ± 0.03 ^ab^

Identical superscripts letters within rows indicate no significant difference (*p* > 0.05); * PM—control sample; PFCDS 5%—bread with 5% hemp flour Dacia Secuieni; PFCDS 10%—bread with 10% hemp flour Dacia Secuieni; PFCDS 15%—bread with 15% hemp flour Dacia Secuieni; PFCDS 20%—bread with 20% hemp flour Dacia Secuieni; PFCZ 5%—bread with 5% hemp flour Zenit; PFCZ 10%—bread with 10% hemp flour Zenit; PFCZ 15%—bread with 15% hemp flour Zenit; PFCZ 20%—bread with 20% hemp flour Zenit. n.d. = not detected.

**Table 6 plants-10-01558-t006:** Pearson correlation for micro and macroelements of bread sample.

Samples	Cu	Cd	Cr	Ni	Pb	Zn	Fe	Mn	Ca	Mg	K	P
PM	−0.579	0.618	−0.576	-	−0.854	−0.940	−0.977	−0.450	−0.847	−0.892	−0.813	−0.439
PFCDS 5%	−0.158	-	−0.166	-	−0.043	−0.011	0.024	−0.297	−0.203	−0.132	−0.229	−0.391
PFCDS 10%	0.042	-	0.207	-	0.068	0.129	0.108	−0.110	0.126	0.090	0.103	0.044
PFCDS 15%	0.308	-	0.356	-	0.208	0.143	0.159	0.224	0.224	0.207	0.179	0.087
PFCDS 20%	0.441	-	0.468	-	0.348	0.311	0.242	0.618	0.305	0.335	0.370	0.541
PFCZ 5%	−0.468	-	−0.390	-	−0.127	−0.102	−0.009	−0.327	−0.237	−0.147	−0.246	−0.446
PFCZ 10%	−0.091	-	−0.203	0.365	0.012	0.102	0.105	−0.228	0.114	0.058	0.099	0.019
PFCZ 15%	0.175	-	0.058	0.480	0.124	0.107	0.144	0.098	0.219	0.185	0.173	0.061
PFCZ 20%	0.330	0.618	0.245	0.595	0.264	0.261	0.205	0.471	0.299	0.295	0.363	0.524

* PM—control sample; PFCDS 5%—bread with 5% hemp flour Dacia Secuieni; PFCDS 10%—bread with 10% hemp flour Dacia Secuieni; PFCDS 15%—bread with 15% hemp flour Dacia Secuieni; PFCDS 20%—bread with 20% hemp flour Dacia Secuieni; PFCZ 5%—bread with 5% hemp flour Zenit; PFCZ 10%—bread with 10% hemp flour Zenit; PFCZ 15%—bread with 15% hemp flour Zenit; PFCZ 20%—bread with 20% hemp flour Zenit.

**Table 7 plants-10-01558-t007:** The content of fatty acids (%) of bread samples.

Samples [% of Total Fatty Acids]	Myristic Acid	Palmitic Acid	Stearic Acid	Oleic Acid	Linoleic Acid	Gamma-Linolenic Acid	Margaric Acid	Palmitoleic Acid	Arachidic Acid	∑ SFA	∑MUFA	∑PUFA
PM	1.88 ± 0.04 ^a^	28.26 ± 0.10 ^a^	3.31 ± 0.03 ^d^	18.34 ± 0.06 ^ab^	42.68 ± 0.10 ^cde^	3.79 ± 0.04 ^e^	n.d.	1.01 ± 0.02 ^a^	0.48 ± 0.03 ^ab^	33.93 ± 0.39 ^a^	19.35 ± 0.07 ^a^	46.47 ± 0.05 ^b^
PFCDS 5%	1.44 ± 0.03 ^c^	22.79 ± 0.13 ^b^	3.91 ± 0.06 ^cd^	17.83 ± 0.08 ^c^	45.73 ± 0.05 ^a^	5.61 ± 0.05 ^cde^	0.85 ± 0.04 ^a^	0.65 ± 0.07 ^b^	0.28 ± 0.03 ^c^	29.27 ± 0.12 ^b^	18.48 ± 0.07 ^bcd^	51.34 ± 0.03 ^a^
PFCDS 10%	1.42 ± 0.04 ^c^	21.36 ± 0.05 ^bc^	4.11 ± 0.02 ^bc^	18.02 ± 0.13 ^bc^	44.47 ± 0.06 ^abc^	6.14 ± 0.03 ^bcd^	0.94 ± 0.03 ^a^	0.57 ± 0.04 ^bc^	0.39 ± 0.06 ^bc^	28.22 ± 0.09 ^b^	18.59 ± 0.07 ^bcd^	50.61 ± 0.06 ^a^
PFCDS 15%	1.40 ± 0.03 ^c^	20.66 ± 0.04 ^bc^	4.45 ± 0.07 ^bc^	18.28 ± 0.10 ^ab^	43.12 ± 0.04 ^bcde^	7.66 ± 0.08 ^ab^	0.89 ± 0.05 ^a^	0.5 ± 0.04 ^bc^	0.4 ± 0.03 ^bc^	27.80 ± 0.15 ^b^	18.78 ± 0.10 ^bc^	50.78 ± 0.05 ^a^
PFCDS 20%	1.34 ± 0.06 ^c^	19.98 ± 0.10 ^bc^	4.64 ± 0.03 ^ab^	18.51 ± 0.03 ^a^	42.53 ± 0.04 ^de^	8.18 ± 0.06 ^a^	0.76 ± 0.03 ^a^	0.41 ± 0.03 ^c^	0.51 ± 0.03 ^ab^	27.23 ± 0.15 ^b^	18.92 ± 0.07 ^ab^	50.71 ± 0.05 ^a^
PFCZ 5%	1.78 ± 0.05 ^ab^	22.03 ± 0.09 ^bc^	3.97 ± 0.10 ^c^	17.71 ± 0.04 ^c^	44.63 ± 0.08 ^ab^	5.07 ± 0.03 ^de^	0.53 ± 0.04 ^a^	0.52 ± 0.04 ^bc^	0.29 ± 0.05 ^c^	28.60 ± 0.12 ^b^	18.23 ± 0.06 ^d^	49.70 ± 0.15 ^a^
PFCZ 10%	1.63 ± 0.05 ^abc^	21.57 ± 0.09 ^bc^	4.06 ± 0.03 ^bc^	17.78 ± 0.04 ^c^	43.48 ± 0.07 ^bcd^	6.73 ± 0.06 ^abcd^	0.73 ± 0.03 ^a^	0.46 ± 0.05 ^bc^	0.39 ± 0.02 ^bc^	28.38 ± 0.23 ^b^	18.24 ± 0.03 ^d^	50.21 ± 0.07 ^a^
PFCZ 15%	1.51 ± 0.03 ^bc^	22.79 ± 0.13 ^b^	4.3 ± 0.03 ^bc^	17.84 ± 0.03 ^c^	42.88 ± 0.05 ^bcde^	7.39 ± 0.06 ^abc^	0.79 ± 0.04 ^a^	0.44 ± 0.06 ^c^	0.46 ± 0.02 ^ab^	29.85 ± 0.11 ^b^	18.28 ± 0.04 ^d^	50.27 ± 0.05 ^a^
PFCZ 20%	1.44 ± 0.05 ^c^	19.31 ± 0.04 ^c^	5.14 ± 0.03 ^a^	17.99 ± 0.03 ^bc^	41.55 ± 0.03 ^e^	8.49 ± 0.08 ^a^	0.86 ± 0.02 ^a^	0.39 ± 0.04 ^c^	0.62 ± 0.04 ^a^	27.37 ± 0.12 ^b^	18.38 ± 0.05 ^cd^	50.04 ± 0.09 ^a^

Identical superscripts letters within rows indicate no significant difference (*p* > 0.05); * PM—control sample; PFCDS 5%—bread with 5% hemp flour Dacia Secuieni; PFCDS 10%—bread with 10% hemp flour Dacia Secuieni; PFCDS 15%—bread with 15% hemp flour Dacia Secuieni; PFCDS 20%—bread with 20% hemp flour Dacia Secuieni; PFCZ 5%—bread with 5% hemp flour Zenit; PFCZ 10%—bread with 10% hemp flour Zenit; PFCZ 15%—bread with 15% hemp flour Zenit; PFCZ 20%—bread with 20% hemp flour Zenit. n.d. = not detected.

**Table 8 plants-10-01558-t008:** Pearson correlation for fatty acids of bread samples.

Samples	Myristic Acid	Palmitic Acid	Stearic Acid	Oleic Acid	Linoleic Acid	Gamma-Linolenic Acid	Margaric Acid	Palmitoleic Acid	Arachidic Acid	∑ SFA	∑MUFA	∑PUFA
PM	0.690	0.891	−0.659	0.409	−0.225	−0.670	-	0.905	0.184	0.908	0.758	−0.941
PFCDS 5%	−0.204	0.101	−0.223	−0.276	0.667	−0.229	0.188	0.200	−0.503	0.056	−0.102	0.352
PFCDS 10%	−0.234	−0.105	−0.071	−0.009	0.229	−0.103	0.303	0.045	−0.116	−0.135	0.007	0.158
PFCDS 15%	−0.281	−0.206	0.176	0.327	−0.098	0.265	0.232	−0.109	−0.078	−0.212	0.194	0.203
PFCDS 20%	−0.403	−0.303	0.312	0.635	−0.270	0.392	0.066	−0.273	0.309	−0.316	0.333	0.185
PFCZ 5%	0.485	−0.008	−0.174	−0.436	0.344	−0.361	−0.226	−0.067	−0.472	−0.066	−0.349	−0.083
PFCZ 10%	0.193	−0.074	−0.107	−0.335	0.007	0.040	0.034	−0.172	−0.127	−0.106	−0.339	0.052
PFCZ 15%	−0.058	0.102	0.066	−0.257	−0.168	0.200	0.113	−0.221	0.117	0.162	−0.0300	0.068
PFCZ 20%	−0.188	−0.399	0.680	−0.057	−0.557	0.466	0.203	−0.308	0.686	−0.291	−0.201	0.007

* PM—control sample; PFCDS 5%—bread with 5% hemp flour Dacia Secuieni; PFCDS 10%—bread with 10% hemp flour Dacia Secuieni; PFCDS 15%—bread with 15% hemp flour Dacia Secuieni; PFCDS 20%—bread with 20% hemp flour Dacia Secuieni; PFCZ 5%—bread with 5% hemp flour Zenit; PFCZ 10%—bread with 10% hemp flour Zenit; PFCZ 15%—bread with 15% hemp flour Zenit; PFCZ 20%—bread with 20% hemp flour Zenit.

**Table 9 plants-10-01558-t009:** The content of amino acids (g/100 g) of bread samples.

Samples [g/100 g]	Arginine	Lysine	Histidine	Phenylalanine	Aspartic Acid	Glutamic Acid	Cysteine	Tyrosine
PM	0.084 ± 0.005 ^b^	0.003 ± 0.002 ^b^	n.d	0.002 ± 0.001 ^b^	n.d.	n.d.	n.d.	0.004 ± 0.002 ^c^
PFCDS 5%	0.102 ± 0.002 ^b^	0.011 ± 0.002 ^b^	0.005 ± 0.001 ^cd^	0.009 ± 0.002 ^b^	0.209 ± 0.002 ^ab^	0.408 ± 0.003 ^bc^	0.008 ± 0.003 ^cd^	0.011 ± 0.002 ^bc^
PFCDS 10%	0.160 ± 0.005 ^b^	0.052 ± 0.003 ^b^	0.009 ± 0.003 ^bcd^	0.500 ± 0.030 ^ab^	0.223 ± 0.115 ^bc^	0.640 ± 0.001 ^abc^	0.013 ± 0.003 ^bcd^	0.027 ± 0.002 ^ab^
PFCDS 15%	0.182 ± 0.005 ^b^	0.098 ± 0.002 ^b^	0.024 ± 0.005 ^ab^	0.970 ± 0.020 ^a^	0.240 ± 0.003 ^a^	0.728 ± 0.002 ^ab^	0.022 ± 0.007 ^abc^	0.035 ± 0.003 ^a^
PFCDS 20%	0.193 ± 0.003 ^b^	0.105 ± 0.003 ^b^	0.034 ± 0.002 ^a^	0.104 ± 0.003 ^b^	0.254 ± 0.116 ^abc^	0.772 ± 0.001 ^a^	0.031 ± 0.006 ^a^	0.042 ± 0.001 ^a^
PFCZ 5%	0.930 ± 0.050 ^a^	0.008 ± 0.001 ^b^	0.003 ± 0.001 ^cd^	0.006 ± 0.003 ^b^	0.207 ± 0.004 ^ab^	0.372 ± 0.004 ^c^	0.007 ± 0.002 ^cd^	0.009 ± 0.003 ^bc^
PFCZ 10%	0.152 ± 0.003 ^b^	0.045 ± 0.004 ^b^	0.007 ± 0.002 ^bcd^	0.043 ± 0.001 ^b^	0.219 ± 0.003 ^a^	0.608 ± 0.003 ^abc^	0.011 ± 0.002 ^cd^	0.024 ± 0.002 ^ab^
PFCZ 15%	0.175 ± 0.004 ^b^	0.093 ± 0.003 ^b^	0.021 ± 0.003 ^abc^	0.091 ± 0.003 ^b^	0.236 ± 0.003 ^a^	0.700 ± 0.004 ^ab^	0.019 ± 0.004 ^abc^	0.033 ± 0.004 ^a^
PFCZ 20%	0.184 ± 0.002 ^b^	0.101 ± 0.005 ^a^	0.029 ± 0.002 ^a^	0.100 ± 0.005 ^ab^	0.248 ± 0.004 ^a^	0.736 ± 0.003 ^a^	0.028 ± 0.004 ^ab^	0.041 ± 0.002 ^a^

Identical superscripts letters within rows indicate no significant difference (*p* > 0.05); * PM—control sample; PFCDS 5%—bread with 5% hemp flour Dacia Secuieni; PFCDS 10%—bread with 10% hemp flour Dacia Secuieni; PFCDS 15%—bread with 15% hemp flour Dacia Secuieni; PFCDS 20%—bread with 20% hemp flour Dacia Secuieni; PFCZ 5%—bread with 5% hemp flour Zenit; PFCZ 10%—bread with 10% hemp flour Zenit; PFCZ 15%—bread with 15% hemp flour Zenit; PFCZ 20%—bread with 20% hemp flour Zenit. n.d. = not detected.

**Table 10 plants-10-01558-t010:** Pearson correlation for amino acids of bread samples.

Samples	Arginine	Lysine	Histidine	Phenylalanine	Aspartic Acid	Glutamic Acid	Cysteine	Tyrosine
PM	−0.225	−0.172	−0.435	−0.243	−0.671	−0.824	−0.534	−0.552
PFCDS 5%	−0.198	−0.155	−0.287	−0.235	0.132	−0.214	−0.257	−0.369
PFCDS 10%	−0.115	−0.075	−0.168	0.275	−0.326	0.132	−0.084	0.049
PFCDS 15%	−0.083	0.016	0.277	0.764	0.251	0.264	0.227	0.259
PFCDS 20%	−0.067	0.029	0.573	−0.137	−0.206	0.329	0.538	0.442
PFCZ 5%	0.988	−0.162	−0.346	−0.239	0.126	−0.268	−0.292	−0.422
PFCZ 10%	−0.126	−0.089	−0.227	−0.200	0.172	0.084	−0.154	−0.029
PFCZ 15%	−0.093	0.006	0.188	−0.150	0.237	0.222	0.123	0.206
PFCZ 20%	−0.080	0.602	0.425	0.165	0.284	0.276	0.434	0.416

* PM—control sample; PFCDS 5%—bread with 5% hemp flour Dacia Secuieni; PFCDS 10%—bread with 10% hemp flour Dacia Secuieni; PFCDS 15%—bread with 15% hemp flour Dacia Secuieni; PFCDS 20%—bread with 20% hemp flour Dacia Secuieni; PFCZ 5%—bread with 5% hemp flour Zenit; PFCZ 10%—bread with 10% hemp flour Zenit; PFCZ 15%—bread with 15% hemp flour Zenit; PFCZ 20%—bread with 20% hemp flour Zenit.

**Table 11 plants-10-01558-t011:** The content of carbohydrate (g/100 g) of bread samples.

Samples [g/100 g]	Sucrose	Fructose	Glucose
PM	9.84 ± 0.02 ^a^	0.44 ± 0.02 ^a^	n.d.
PFCDS 5%	8.53 ± 0.01 ^ab^	0.42 ± 0.03 ^a^	n.d.
PFCDS 10%	7.84 ± 0.02 ^b^	0.37 ± 0.01 ^abc^	n.d.
PFCDS 15%	6.92 ± 0.04 ^bc^	0.32 ± 0.02 ^bcde^	n.d.
PFCDS 20%	5.25 ± 0.02 ^c^	0.28 ± 0.03 ^de^	n.d.
PFCZ 5%	8.47 ± 0.03 ^ab^	0.39 ± 0.05 ^ab^	n.d.
PFCZ 10%	7.62 ± 0.01 ^b^	0.34 ± 0.03 ^bcd^	n.d.
PFCZ 15%	6.78 ± 0.03 ^bc^	0.30 ± 0.03 ^cde^	n.d.
PFCZ 20%	5.17 ± 0.02 ^c^	0.25 ± 0.02 ^e^	n.d.

Identical superscripts letters within rows indicate no significant difference (*p* > 0.05); * PM—control sample; PFCDS 5%—bread with 5% hemp flour Dacia Secuieni; PFCDS 10%—bread with 10% hemp flour Dacia Secuieni; PFCDS 15%—bread with 15% hemp flour Dacia Secuieni; PFCDS 20%—bread with 20% hemp flour Dacia Secuieni; PFCZ 5%—bread with 5% hemp flour Zenit; PFCZ 10%—bread with 10% hemp flour Zenit; PFCZ 15%—bread with 15% hemp flour Zenit; PFCZ 20%—bread with 20% hemp flour Zenit. n.d. = not detected.

**Table 12 plants-10-01558-t012:** Pearson correlation for carbohydrate of bread samples.

Samples	Sucrose	Fructose
PM	0.843	0.513
PFCDS 5%	0.134	0.404
PFCDS 10%	0.021	0.133
PFCDS 15%	−0.130	−0.139
PFCDS 20%	−0.405	−0.356
PFCZ 5%	0.124	0.241
PFCZ 10%	−0.015	−0.030
PFCZ 15%	−0.154	−0.247
PFCZ 20%	−0.418	−0.519

* PM—control sample; PFCDS 5%—bread with 5% hemp flour Dacia Secuieni; PFCDS 10%—bread with 10% hemp flour Dacia Secuieni; PFCDS 15%—bread with 15% hemp flour Dacia Secuieni; PFCDS 20%—bread with 20% hemp flour Dacia Secuieni; PFCZ 5%—bread with 5% hemp flour Zenit; PFCZ 10%—bread with 10% hemp flour Zenit; PFCZ 15%—bread with 15% hemp flour Zenit; PFCZ 20%—bread with 20% hemp flour Zenit.

**Table 13 plants-10-01558-t013:** Microbiological parameters (total combined yeasts and molds- TYMC) (ufc/g) of bread samples.

Samples	TYMC [ufc/g] in Day 1	TYMC [ufc/g] in Day 2	TYMC [ufc/g] in Day 3
PM %	47 ± 1.00 ^a^	45 ± 2.00 ^a^	42 ± 1.00 ^a^
PFCDS 5%	34 ± 2.00 ^b^	32 ± 1.00 ^b^	30 ± 2.00 ^b^
PFCDS 10%	28 ± 1.00 ^bc^	27 ± 2.00 ^bc^	24 ± 1.00 ^bc^
PFCDS 15%	26 ± 3.00 ^cd^	25 ± 3.00 ^cd^	22 ± 1.00 ^c^
PFCDS 20%	22 ± 2.00 ^cd^	20 ± 2.00 ^de^	20 ± 1.00 ^cd^
PFCZ 5%	24 ± 1.00 ^cd^	22 ± 1.00 ^cde^	20 ± 3.00 ^cd^
PFCZ 10%	21 ± 3.00 ^cde^	20 ± 2.00 ^de^	19 ± 2.00 ^cd^
PFCZ 15%	19 ± 2.00 ^de^	17 ± 2.00 ^ef^	15 ± 3.00 ^de^
PFCZ 20%	14 ± 1.00 ^e^	12 ± 1.00 ^f^	10 ± 1.00 ^e^

Identical superscripts letters within rows indicate no significant difference (*p* > 0.05); * PM—control sample; PFCDS 5%—bread with 5% hemp flour Dacia Secuieni; PFCDS 10%—bread with 10% hemp flour Dacia Secuieni; PFCDS 15%—bread with 15% hemp flour Dacia Secuieni; PFCDS 20%—bread with 20% hemp flour Dacia Secuieni; PFCZ 5%—bread with 5% hemp flour Zenit; PFCZ 10%—bread with 10% hemp flour Zenit; PFCZ 15%—bread with 15% hemp flour Zenit; PFCZ 20%—bread with 20% hemp flour Zenit.

**Table 14 plants-10-01558-t014:** Pearson corelation between moisture and microbiology content of bread samples.

Days	Moisture + Microbiology
Day 1	0.953
Day 2	0.945
Day 3	0.944

**Table 15 plants-10-01558-t015:** Pearson correlation for microbiological parameters (total combined yeasts and molds) of bread samples.

Samples	Day 1	Day 2	Day 3
PM	0.799	0.789	0.786
PFCDS 5%	0.302	0.290	0.304
PFCDS 10%	0.072	0.098	0.063
PFCDS 15%	−0.004	0.021	−0.018
PFCDS 20%	−0.157	−0.171	−0.098
PFCZ 5%	−0.081	−0.094	−0.098
PFCZ 10%	−0.195	−0.171	−0.138
PFCZ 15%	−0.272	−0.286	−0.299
PFCZ 20%	−0.463	−0.478	−0.500

* PM—control sample; PFCDS 5%—bread with 5% hemp flour Dacia Secuieni; PFCDS 10%—bread with 10% hemp flour Dacia Secuieni; PFCDS 15%—bread with 15% hemp flour Dacia Secuieni; PFCDS 20%—bread with 20% hemp flour Dacia Secuieni; PFCZ 5%—bread with 5% hemp flour Zenit; PFCZ 10%—bread with 10% hemp flour Zenit; PFCZ 15%—bread with 15% hemp flour Zenit; PFCZ 20%—bread with 20% hemp flour Zenit.

**Table 16 plants-10-01558-t016:** Texture profile analyses content of bread samples.

Samples	Hardness Cycle 1 [g]	Total Work Cycle 1 [mj]	Hardness Cycle 2 [g]	Cohesiveness [n.a.]	Total Work Cycle 2 [mj]	Springiness Index [n.a.]	Viscosity [g]	Chewiness [mj]
PM	759 ± 3.51 ^abc^	42.2 ± 0.11 ^b^	702.67 ± 0.22 ^ab^	0.69 ± 0.09 ^a^	30.87 ± 0.10 ^a^	0.93 ± 0.03 ^b^	520.33 ± 0.11 ^ab^	41.6 ± 0.08 ^ab^
PFCDS 5%	910.33 ± 0.32 ^a^	46.13 ± 0.05 ^ab^	850 ± 7.51 ^a^	0.67 ± 0.15 ^a^	31.97 ± 0.06 ^a^	0.92 ± 0.03 ^b^	612.67 ± 0.08 ^a^	47.57 ± 0.09 ^a^
PFCDS 10%	417.33 ± 0.07 ^d^	21.1 ± 0.04 ^b^	390.67 ± 0.16 ^d^	0.69 ± 0.03 ^a^	15.5 ± 0.11 ^b^	0.89 ± 0.03 ^b^	221.67 ± 0.08 ^d^	22.23 ± 0.09 ^c^
PFCDS 15%	566 ± 3.00 ^bcd^	26.17 ± 0.04 ^b^	522 ± 4.51 ^bcd^	0.64 ± 0.05 ^a^	17.7 ± 0.10 ^b^	0.89 ± 0.03 ^b^	358 ± 3.51 ^bcd^	25.23 ± 0.09 ^c^
PFCDS 20%	503.67 ± 0.15 ^cd^	30.03 ± 0.03 ^b^	456.33 ± 0.14 ^cd^	0.62 ± 0.03 ^a^	20 ± 2.52 ^b^	0.90 ± 0.04 ^b^	313 ± 3.51 ^cd^	25.67 ± 0.07 ^c^
PFCZ 5%	639.67 ± 0.08 ^bcd^	29.07 ± 0.03 ^b^	589.67 ± 0.16 ^bc^	0.68 ± 0.08 ^a^	21.07 ± 0.03 ^b^	0.92 ± 0.03 ^b^	433.33 ± 0.10 ^bc^	32.3 ± 0.08 ^bc^
PFCZ 10%	796.33 ± 0.09 ^ab^	70.77 ± 0.07 ^a^	504.67 ± 0.13 ^cd^	0.43 ± 0.02 ^a^	20 ± 2.00 ^b^	1.11 ± 0.04 ^a^	326.33 ± 0.09 ^cd^	25.7 ± 0.06 ^c^
PFCZ 15%	514 ± 3.51 ^cd^	26 ± 3.51 ^b^	469 ± 4.51 ^cd^	0.67 ± 0.05 ^b^	18.37 ± 0.08 ^b^	0.91 ± 0.03 ^b^	343.67 ± 0.08 ^cd^	25.97 ± 0.08 ^c^
PFCZ 20%	387 ± 2.52 ^d^	20 ± 2.52 ^b^	443.67 ± 0.16 ^cd^	0.64 ± 0.03 ^a^	13.93 ± 0.06 ^b^	0.89 ± 0.04 ^b^	310.33 ± 0.10 ^cd^	21.27 ± 0.11 ^c^

Identical superscripts letters within rows indicate no significant difference (*p* > 0.05); * PM—control sample; PFCDS 5%—bread with 5% hemp flour Dacia Secuieni; PFCDS 10%—bread with 10% hemp flour Dacia Secuieni; PFCDS 15%—bread with 15% hemp flour Dacia Secuieni; PFCDS 20%—bread with 20% hemp flour Dacia Secuieni; PFCZ 5%—bread with 5% hemp flour Zenit; PFCZ 10%—bread with 10% hemp flour Zenit; PFCZ 15%—bread with 15% hemp flour Zenit; PFCZ 20%—bread with 20% hemp flour Zenit.

**Table 17 plants-10-01558-t017:** Pearson correlation for texture profile analyses content of bread samples.

Samples	Hardness Cycle 1 [g]	Total Work Cycle 1 [mj]	Hardness Cycle 2 [g]	Cohesiveness [n.a.]	Total Work Cycle 2 [mj]	Springiness Index [n.a.]	Viscosity [g]	Chewiness [mj]
PM	0.311	0.176	0.399	0.246	0.585	0.006	0.431	0.490
PFCDS 5%	0.627	0.267	0.779	0.154	0.650	−0.048	0.719	0.736
PFCDS 10%	−0.404	−0.313	−0.404	0.246	−0.330	−0.210	−0.500	−0.309
PFCDS 15%	−0.093	−0.196	−0.066	0.015	−0.199	−0.210	−0.075	−0.186
PFCDS 20%	−0.223	−0.106	−0.235	−0.077	−0.062	−0.156	−0.216	−0.167
PFCZ 5%	0.061	−0.128	0.108	0.200	0.001	−0.048	0.160	0.106
PFCZ 10%	0.389	0.839	−0.111	−0.954	−0.062	0.977	−0.174	−0.166
PFCZ 15%	−0.202	−0.200	−0.202	0.154	−0.159	−0.102	−0.120	−0.155
PFCZ 20%	−0.467	−0.339	−0.268	0.015	−0.423	−0.210	−0.224	−0.349

* PM—control sample; PFCDS 5%—bread with 5% hemp flour Dacia Secuieni; PFCDS 10%—bread with 10% hemp flour Dacia Secuieni; PFCDS 15%—bread with 15% hemp flour Dacia Secuieni; PFCDS 20%—bread with 20% hemp flour Dacia Secuieni; PFCZ 5%—bread with 5% hemp flour Zenit; PFCZ 10%—bread with 10% hemp flour Zenit; PFCZ 15%—bread with 15% hemp flour Zenit; PFCZ 20%—bread with 20% hemp flour Zenit.

**Table 18 plants-10-01558-t018:** The recipe for classic bread and bread enriched with hemp flour.

Raw and Auxiliary Materials	Bread with Hemp Flour Variety Dacia Secuieni and Zenit with Wheat Flour					
	Wheat Flour Type 550 (kg)	Hemp Flour (kg)		Yeast (kg)	Salt (kg)	Water (l)
PM	0.8	-		0.02	0.0145	0.416
PFCDS 5%	0.76	0.04		0.02	0.0145	0.416
PFCDS 10%	0.72	0.08		0.02	0.0145	0.416
PFCDS 15%	0.68	0.12		0.02	0.0145	0.416
PFCDS 20%	0.64	0.16		0.02	0.0145	0.416
PFCZ 5%	0.76	0.04		0.02	0.0145	0.416
PFCZ 10%	0.72	0.08		0.02	0.0145	0.416
PFCZ 15%	0.68	0.12		0.02	0.0145	0.416
PFCZ 20%	0.64	0.16		0.02	0.0145	0.416
**Technological parameters**						
Parameter name			Minute			
Fermentation			60			
Rising			60			
Final Rising			30			
Baking time			45			

* PM—control sample; PFCDS 5%—bread with 5% hemp flour Dacia Secuieni; PFCDS 10%—bread with 10% hemp flour Dacia Secuieni; PFCDS 15%—bread with 15% hemp flour Dacia Secuieni; PFCDS 20%—bread with 20% hemp flour Dacia Secuieni; PFCZ 5%—bread with 5% hemp flour Zenit; PFCZ 10%—bread with 10% hemp flour Zenit; PFCZ 15%—bread with 15% hemp flour Zenit; PFCZ 20%—bread with 20% hemp flour Zenit.

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
