# Peer review of "Hemp (Cannabis sativa L.) Flour-Based Wheat Bread as Fortified Bakery Product"

_plants, 2021, doi:10.3390/plants10081558_

Round 1

Reviewer 1 Report

I will give all comments in order by line number:

Line 34 – "8 amino acids" – do not start the sentence with a digit but with the word --> "Eight amino acids"

Line 35 – "… / 100g" – the space between the number and the unit of measurement is missing --> "… / 100 g"

Line 56 – "the ex-istente"– spelling mistake --> "existent"

Line 69 – "non-trient" – spelling mistake --> "nutrients"

Line 69 – "non-trient and vitamins" – vitamins are nutrients as well, so just nutrients --> "nutrients"

Line 72 – "…, More" – use a lowercase letter after the comma --> "more"

Lines 69, 498, 501 – "Cannabis sativa L." – genus and species should be written in italics --> "Cannabis sativa L."

Line 86 – "in this regard These" – the dot at the end of the sentence is missing --> "in this regard. These"

Line 96 – "This report is essential…" – please rephrase; it is not the report that is essential for the proper functioning of the body etc.

Line 99 – "edestin (a legume)" – please clarify: legumin class reserve protein and a seed storage protein… and edestin is also globular protein

Lines 101, 543, 545, 552, 568, 608 – "° C" --> "°C"

Line 105 – "reducing … obesity, and diabetes" – please clarify: obesity and diabetes rate? risk? complications?

Lines 108 and 110 – "are" "were"– please match the verb tenses in the same sentence

Line 134 – "respectively" – respectively to what? please complete the sentence. Please check other sentences where you use adverb respectively.

Lines 151, 162, 169, 175, 185 – "Increases" – you need to use a noun, not 3rd person present (verb) --> "Increase"

Line 159 – "reports increases" --> "report increase"

Line 175 – "which" --> "who"

Tables 1, 5, 7, 14 – When superscript letters move to a new row the table loses clarity

Line 219 – "reports decreases" --> "report decrease"

Line 245 – "and K" --> "and of K"

Line 251 – "reports" --> "report"

Line 252-253 – "identifies" – you need to use (3rd person plural) past tense, not 3rd person singular present --> "identified"

Line 253 – "shows" --> "research shows"

Line 277 – "9 saturated" – do not start the sentence with a digit but with the word --> "Nine saturated"

Line 278 – "Saturated fatty acids are" – use the same tense as in the previous sentence to which you refer --> "Saturated fatty acids were"

Line 281 – "a little" – ? please specify

Line 289 – "ga-ma" --> "gamma"

Lines 293, 313, 316 – "γ" – please use gamma or γ throughout the text

Line 293 – "18: 2" --> "18:2"

Line 293 – "18: 3" --> "18:3"

Line 297-298 – "oppressive source" – I am not sure about the word oppressive, maybe you should find/choose term among Permitted nutrition claims

Line 303 – "Which reports" --> "who reported"

Line 309 – "pal-mitoleic" --> "palmitoleic"

Line 336 – "fortified plant products" --> "plant fortified products"

Line 336-337 – "with plant sources is increasingly being promoted by nutritionists " – please rephrase

Line 349 – "in all" --> "for all"

Line 350 – "increases" --> "increase"

Line 375-376 – "is the date of sucrose" – date is a completely wrong word --> I suggest "refers to sucrose"

Line 379 – " The main cause is due to the process of baking bread" --> " The main cause is the process of baking bread"

Line 393 – "date" – date is a completely wrong word meaning the day of the month or year as specified by a number --> "data"

Line 465 – "witness bread" – ? --> "control sample bread"

Line 465 – "P.F.C. S.D. 10%" – please use the same abbreviations throughout the text --> "PFCDS 10%"

Line 467 – "P.F.C. S.D. 5%" – please use the same abbreviations throughout the text --> "PFCDS 5%"

Line 467 – "P.F.C. S.D. 10%" – please use the same abbreviations throughout the text --> "PFCDS 10%"

Line 471 – "P.F.C. S.D. 5%" – please use the same abbreviations throughout the text --> "PFCDS 5%"

Line 472 – "P.F.C. Z. 20%" – please use the same abbreviations throughout the text --> "PFCZ 20%"

Line 475 – "are also" --> "were also"

Line 479 – "re-marked" --> "remarked"

Lines 536-537 – a blank line is missing between sections 3.3. and 3.4.

Line 541 – "this takes" --> "that takes"

Lines 542, 546 – "2mL" --> "2 mL"

Line 553 – "0C" --> " °C"

Line 580-581 – "Weigh 3 g of the sample and grind it well, then add 8 ml of extraction solvent water." – please rephrase, use passive voice

Line 596 – "ml" – use the same way of writing mL as through the previous text --> "mL"

Line 601 – "millilitres" – misspelled --> "mL"

Line 652 – "of fortification in bread making" --> "of nutrients in fortified bread making"

Author Response

The Editor

Plants

Subject: Submission of revised manuscript No. Plants: 1299515

Dear Sir

It is stated that I want to submit revised article entitled, “Hemp (Cannabis sativa L.) flour-based wheat bread as fortified bakery

product” for publication in your esteemed Journal. We are highly thankful to referees whose comments helped in improving this manuscript. We have revised the entire manuscript for language as well as for proper flow of the information. Mostly these are the reviewers’ observations which are addressed in the point by point rebuttal file and also incorporated the same in the text.

Below is response to reviewers comments:

Reviewer 1

Comment: I will give all comments in order by line number:

Authors response: Thank you for all encouraging comments.

Comment:Line 34 – "8 amino acids" – do not start the sentence with a digit but with the word --> "Eight amino acids"

Authors response: corrected

Comment:Line 35 – "… / 100g" – the space between the number and the unit of measurement is missing --> "… / 100 g"

Authors response: corrected

Comment:Line 56 – "the ex-istente"– spelling mistake --> "existent"

Authors response: corrected

Comment:Line 69 – "non-trient" – spelling mistake --> "nutrients"

Authors response: corrected

Comment:Line 69 – "non-trient and vitamins" – vitamins are nutrients as well, so just nutrients --> "nutrients"

Authors response: corrected

Comment:Line 72 – "…, More" – use a lowercase letter after the comma --> "more"

Authors response: corrected

Comment:Lines 69, 498, 501 – "Cannabis sativa L." – genus and species should be written in italics --> "Cannabis sativa L."

Authors response: corrected

Comment:Line 86 – "in this regard These" – the dot at the end of the sentence is missing --> "in this regard. These"

Authors response: corrected

Comment:Line 96 – "This report is essential…" – please rephrase; it is not the report that is essential for the proper functioning of the body etc.

Authors response: corrected. (Thus, hemp seed oils help the proper functioning of the body and prevent neurodegenerative diseases, cardiovascular and various types of cancer.)

Comment: Line 99 – "edestin (a legume)" – please clarify: legumin class reserve protein and a seed storage protein… and edestin is also globular protein

Authors response: corrected. (The predominant proteins are edestin, a storage protein albumin (a globular protein).

Comment:Lines 101, 543, 545, 552, 568, 608 – "° C" --> "°C"

Authors response: corrected

Comment:Line 105 – "reducing … obesity, and diabetes" – please clarify: obesity and diabetes rate? risk? complications?

Authors response: corrected. (risk of obesity and diabetes)

Comment:Lines 108 and 110 – "are" "were"– please match the verb tenses in the same sentence

Authors response: corrected. (are)

Comment:Line 134 – "respectively" – respectively to what? please complete the sentence. Please check other sentences where you use adverb respectively.

Authors response: corrected. (This decrease is due to the low humidity of the hemp flour used 4.49% for hemp flour variety Dacia Secuieni, respectively 4.98% for hemp flour variety Zenit).

Comment:Lines 151, 162, 169, 175, 185 – "Increases" – you need to use a noun, not 3rd person present (verb) --> "Increase"

Authors response: corrected

Comment:Line 159 – "reports increases" --> "report increase"

Authors response: corrected

Comment:Line 175 – "which" --> "who"

Authors response: corrected

Comment:Tables 1, 5, 7, 14 – When superscript letters move to a new row the table loses clarity

Authors response: corrected

Comment:Line 219 – "reports decreases" --> "report decrease"

Authors response: corrected

Comment:Line 245 – "and K" --> "and of K"

Authors response: corrected

Comment:Line 251 – "reports" --> "report"

Authors response: corrected

Comment:Line 252-253 – "identifies" – you need to use (3rd person plural) past tense, not 3rd person singular present --> "identified"

Authors response: corrected

Comment:Line 253 – "shows" --> "research shows"

Authors response: corrected

Comment:Line 277 – "9 saturated" – do not start the sentence with a digit but with the word --> "Nine saturated"

Authors response: corrected

Comment:Line 278 – "Saturated fatty acids are" – use the same tense as in the previous sentence to which you refer --> "Saturated fatty acids were"

Authors response: corrected

Comment:Line 281 – "a little" – ? please specify

Authors response: corrected. (It can be seen that it decreases with the addition of hemp flour)

Comment:Line 289 – "ga-ma" --> "gamma"

Authors response: corrected

Comment:Lines 293, 313, 316 – "γ" – please use gamma or γ throughout the text

Authors response: corrected

Comment:Line 293 – "18: 2" --> "18:2"

Authors response: corrected

Comment:Line 293 – "18: 3" --> "18:3"

Authors response: corrected

Comment:Line 297-298 – "oppressive source" – I am not sure about the word oppressive, maybe you should find/choose term among Permitted nutrition claims

Authors response: corrected. Consequently, bread with the addition of hemp seed flour is considered an adequate source of EFA.

Comment:Line 303 – "Which reports" --> "who reported"

Authors response: corrected

Comment:Line 309 – "pal-mitoleic" --> "palmitoleic"

Authors response: corrected

Comment:Line 336 – "fortified plant products" --> "plant fortified products"

Authors response: corrected

Comment:Line 336-337 – "with plant sources is increasingly being promoted by nutritionists " – please rephrase

Authors response: corrected. Due to the promotion by nutritionists of the quality of vegetable proteins, plant fortified products are in great demand.

Comment:Line 349 – "in all" --> "for all"

Authors response: corrected.

Comment:Line 350 – "increases" --> "increase"

Authors response: corrected.

Comment:Line 375-376 – "is the date of sucrose" – date is a completely wrong word --> I suggest "refers to sucrose"

Authors response: corrected.

Comment:Line 379 – " The main cause is due to the process of baking bread" --> " The main cause is the process of baking bread"

Authors response: corrected.

Comment:Line 393 – "date" – date is a completely wrong word meaning the day of the month or year as specified by a number --> "data"

Authors response: corrected.

Comment:Line 465 – "witness bread" – ? --> "control sample bread"

Authors response:

Comment:Line 465 – "P.F.C. S.D. 10%" – please use the same abbreviations throughout the text --> "PFCDS 10%"

Authors response: corrected.

Comment:Line 467 – "P.F.C. S.D. 5%" – please use the same abbreviations throughout the text --> "PFCDS 5%"

Authors response: corrected.

Comment:Line 467 – "P.F.C. S.D. 10%" – please use the same abbreviations throughout the text --> "PFCDS 10%"

Authors response: corrected.

Comment:Line 471 – "P.F.C. S.D. 5%" – please use the same abbreviations throughout the text --> "PFCDS 5%"

Authors response: corrected.

Comment:Line 472 – "P.F.C. Z. 20%" – please use the same abbreviations throughout the text --> "PFCZ 20%"

Authors response: corrected.

Comment:Line 475 – "are also" --> "were also"

Authors response: corrected.

Comment:Line 479 – "re-marked" --> "remarked"

Authors response: corrected.

Comment:Lines 536-537 – a blank line is missing between sections 3.3. and 3.4.

Authors response: corrected.

Comment:Line 541 – "this takes" --> "that takes"

Authors response: corrected.

Comment:Lines 542, 546 – "2mL" --> "2 mL"

Authors response: corrected.

Comment:Line 553 – "0C" --> " °C"

Authors response: corrected.

Comment:Line 580-581 – "Weigh 3 g of the sample and grind it well, then add 8 ml of extraction solvent water." – please rephrase, use passive voice

Authors response: corrected.

Comment:Line 596 – "ml" – use the same way of writing mL as through the previous text --> "mL"

Authors response: corrected.

Comment:Line 601 – "millilitres" – misspelled --> "mL"

Authors response: corrected.

Comment: Line 652 – "of fortification in bread making" --> "of nutrients in fortified bread making"

Authors response: corrected

Reviewer 2

Thank you for all encouraging comments.

Comment: English writing: The text should undergo scientific writing. In many places it is awkward.

Authors response: The entire manuscript has been corrected

Comment: This manuscript presents a very large data set. The data is presented in cumbersome tables, which are hard to follow, with 2 tables for each data set. The authors needs to find a wat to present the data in a format that will be easier to follow. Some possibilities are: to change some of the tables to figure format; to change the first column to the left to reflect the variety tested and the % hemp oil (in a two column format).

Authors response: We tried to make figures and graphs, but because we have a lot of data, they are very crowded and the values for each sample are no longer visible.

I arranged the data in the table to be much easier to track.

Comment: The goal of the project should be spelled out at the beginning of the abstract.

Authors response: The goal is presented in the second sentence of the abstract.

Comment: Abstract line 29: The sentence is awkward, to what does the word 'thus' refer to?

Authors response: corrected

Comment: Line 32: it is not clear what do you mean by "in bread with addition".

Authors response: corrected. (in bread with addition of hemp flour)

Comment: There is a mix of present and past tense in the abstract which should be corrected- for past tense presentation.

Authors response: corrected

Comment: Line 32: the sentence about minerals needs to be extended to reflect results. As it is, it is not connected to the text.

Authors response: corrected

Comment: Line 68-73 : The text needs to be revised for proper presentation of the references. Check also the manuscript throughout.

Authors response: all references have been checked

 Comment: Latin names such as Cannabis sativa should be written in italics.

Authors response: corrected.

Comment: Line 86: Plant female should be female plant.

Authors response: corrected.

Comment: Line 94: "It is very important to remember" is not an appropriate language style for a scientific paper.

Authors response: corrected.

Comment: Line 110: The authors should acknowledge the fact that variation in cultivation conditions and genotypes are known to affect composition on minerals in the cannabis sativa plant organs. I recommend to add a sentence such as this to the end of line 110: "Concentrations of micronutrients and macronutrients in cannabis sativa plants vary between varieties and plant organs (http://dx.doi.org/10.1016/j.indcrop.2018.11.039 ; https://acsess.onlinelibrary.wiley.com/doi/abs/10.2134/agronj1977.00021962006900050026x ) and are affected by cultivation conditions (https://doi.org/10.3389/fpls.2020.572293 ; https://www.notulaebotanicae.ro/index.php/nbha/article/view/11527; https://acsess.onlinelibrary.wiley.com/doi/full/10.2134/cftm2015.0159; https://doi.org/10.3389/fpls.2019.01369 ).

Authors response: added.

Comment: Section 3.3. Determination of micro and macroelements. The link provided connects to a website that requires payment to access the data. Yu need to provide a conventional link to the analytical procedure, or a full description.

Authors response: corrected (I have added a source in which it is detailed)

Comment: Table 3: unites are missing from the table headings.

Authors response: corrected

Comment: Table 5: unites are missing from the table headings.

Authors response: corrected

Comment: Tables 7, 9, 11: unites are missing from the table headings.

Authors response: corrected

Thank you once again for your valuable comments. I am available if there are any further queries.

--

Best regards,

Marc (Vlaic) Romina Alina et al.

Reviewer 2 Report

  1. English writing: The text should undergo scientific writing. In many places it is awkward.
  2. This manuscript presents a very large data set. The data is presented in cumbersome tables, which are hard to follow, with 2 tables for each data set. The authors needs to find a wat to present the data in a format that will be easier to follow. Some possibilities are: to change some of the tables to figure format; to change the first column to the left to reflect the variety tested and the % hemp oil (in a two column format).
  3. The goal of the project should be spelled out at the beginning of the abstract.
  4. Abstract line 29: The sentence is awkward, to what does the word 'thus' refer to?
  5. Line 32: it is not clear what do you mean by "in bread with addition".
  6. There is a mix of present and past tense in the abstract which should be corrected- for past tense presentation.
  7. Line 32: the sentence about minerals needs to be extended to reflect results. As it is, it is not connected to the text.
  8. Line 68-73 : The text needs to be revised for proper presentation of the references. Check also the manuscript throughout.
  9.  Latin names such as Cannabis sativa should be written in italics.
  10. Line 86: Plant female should be female plant.
  11. Line 94: "It is very important to remember" is not an appropriate language style for a scientific paper.
  12. Line 110: The authors should acknowledge the fact that variation in cultivation conditions and genotypes are known to affect composition on minerals in the cannabis sativa plant organs. I recommend to add a sentence such as this to the end of line 110: "Concentrations of micronutrients and macronutrients in cannabis sativa plants vary between varieties and plant organs (http://dx.doi.org/10.1016/j.indcrop.2018.11.039 ; https://acsess.onlinelibrary.wiley.com/doi/abs/10.2134/agronj1977.00021962006900050026x ) and are affected by cultivation conditions (https://doi.org/10.3389/fpls.2020.572293 ; https://www.notulaebotanicae.ro/index.php/nbha/article/view/11527; https://acsess.onlinelibrary.wiley.com/doi/full/10.2134/cftm2015.0159; https://doi.org/10.3389/fpls.2019.01369 ).
  13. Section 3.3. Determination of micro and macroelements. The link provided connects to a website that requires payment to access the data. Yu need to provide a conventional link to the analytical procedure, or a full description.
  14. Table 3: unites are missing from the table headings.
  15. Table 5: unites are missing from the table headings.
  16. Tables 7, 9, 11: unites are missing from the table headings.

Author Response

(The authors gave the same response as above.)

Round 2

Reviewer 2 Report

English writing: The text should undergo scientific writing. In many places, it is awkward. The revisions conducted are not sufficient. I listed below only a few places with issue. There are many more.

At the 1st line of the abstract the authors write that Hemp flower was added to bread. They do not give any details about the bread. This is an issue also at the last section of the introduction.

Line 94-95: there is a built-in contradiction in this sentence. You do not detail 2 phytochemical with psychoactive effects.

The last sentence of the introduction should be in the past tense.

Large parts of the study, including the result section are written in the present tense. This needs to be changed to the past tense.

Line 209: the english needs to be revised. The correction made did not improve the english.

The section starting at line 216: the english needs to be revised.

The next section starting in line 224 is also awkward.

There are issues also in the next section starting in line 228.

There are issues with the references in the text. See for example line 265.

The English in the title of table 3 is awkward. The English should be revised.

Table 4 was not revised. Units are still missing.

Same comment for table 5 and table 6. (Please check all tables).

Table 13: The 1st raw of the table should indicate what was measured.

Figs 14+15 should be combined.

Throughout the manuscript the author refer to a treatment as 'sample'. For example in line 567. This should be revised because not only one sample was measured (replications). Treatment is therefore a better terminology.

Line 696: the style of writing is not appropriate for M&M in a ms. It reads as part of a working protocol.

The authors failed to conduct all recommended changes. For example the goal of the project is still not spelled out at the beginning of the abstract. You can't start an abstract with methodology.

Author Response

The Editor

Plants

Subject: Submission of revised manuscript No. Plants: 1299515

Dear Sir

It is stated that I want to submit revised article entitled, “Hemp (Cannabis sativa L.) flour-based wheat bread as fortified bakery

product” for publication in your esteemed Journal. We are highly thankful to referees whose comments helped in improving this manuscript. We have revised the entire manuscript for language as well as for proper flow of the information. Mostly these are the reviewers’ observations which are addressed in the point by point rebuttal file and also incorporated the same in the text.

Below is response to reviewers comments:

Reviewer 2

Comment: English writing: The text should undergo scientific writing. In many places, it is awkward. The revisions conducted are not sufficient. I listed below only a few places with issue. There are many more.

Authors response: The text was read and corrected by a person with a Cambridge certificate

Comment: At the 1st line of the abstract the authors write that Hemp flower was added to bread. They do not give any details about the bread. This is an issue also at the last section of the introduction.

Authors response: Estimated reviwer all details are given in section 3.1 and all values are presented in Table 18

Comment: Line 94-95: there is a built-in contradiction in this sentence. You do not detail 2 phytochemical with psychoactive effects.

Authors response: corrected

Comment: The last sentence of the introduction should be in the past tense.

Authors response: corrected

Comment: Large parts of the study, including the result section are written in the present tense. This needs to be changed to the past tense.

Authors response: corrected

Comment: Line 209: the english needs to be revised. The correction made did not improve the english.

Authors response: corrected

Comment: The section starting at line 216: the english needs to be revised.

Authors response: corrected

Comment: The next section starting in line 224 is also awkward.

Authors response: corrected

Comment: There are issues also in the next section starting in line 228.

Authors response: corrected

Comment: There are issues with the references in the text. See for example line 265.

Authors response: If you are referring to section 2.2 I have checked all the references and they are correct. Which reference does not correspond in your opinion?

Comment: The English in the title of table 3 is awkward. The English should be revised.

Authors response: corrected

Comment: Table 4 was not revised. Units are still missing.

Authors response: Estimated reviewer Table 4 presents pearson correlation. We do not have units of measurement for these correlations

Comment: Same comment for table 5 and table 6. (Please check all tables).

Authors response: Table 5 shows the unit of measurement (mg / 100 g), and Table 6 shows the pearson correlation. We do not have units of measurement for these correlations

Comment: Table 13: The 1st raw of the table should indicate what was measured.

Authors response: corrected

Comment: Figs 14+15 should be combined.

Authors response: In the manuscript we do not have fig 14 and fig 15.

If you refer to Table 14 and Table 15, the two tables cannot be combined because different pearson correlations are presented. Table 14 shows: Pearson correlation for microbiological parameters (total combined yeasts and molds) of bread samples, and Table 15 Pearson correlation between moisture and microbiology content of bread samples

Comment: Throughout the manuscript the author refer to a treatment as 'sample'. For example in line 567. This should be revised because not only one sample was measured (replications). Treatment is therefore a better terminology.

Authors response: corrected

Comment: Line 696: the style of writing is not appropriate for M&M in a ms. It reads as part of a working protocol.

Authors response: corrected

Comment: The authors failed to conduct all recommended changes. For example the goal of the project is still not spelled out at the beginning of the abstract. You can't start an abstract with methodology.

Authors response: corrected

Thank you once again for your valuable comments. I am available if there are any further queries.

--

Best regards,

Marc (Vlaic) Romina Alina et al.
